# Effects of Plant Growth Regulators on the Rapid Propagation System of *Broussonetia papyrifera* L. Vent Explants

**Jiakang Zhou** [1] , **Yang Liu** [1] , **Liang Wu** [1] , **Yunlin Zhao** [1] , **Wan Zhang** [1] , **Guiyan Yang** [2] **and Zhenggang Xu** [1,2,*]

[1]  Hunan Research Center of Engineering Technology for Utilization of Environmental and Resources Plant, Central South University of Forestry and Technology, Changsha 410004, China; zjk@csuft.edu.cn (J.Z.); lysj408@csuft.edu.cn (Y.L.); wuliang9304@163.com (L.W.); zyl8291290@163.com (Y.Z.); zw@csuft.edu.cn (W.Z.)

[2]  Key Laboratory of National Forestry and Grassland Administration on Management of Western Forest Bio-Disaster, College of Forestry, Northwest A&F University, Yangling 712100, China; yangguiyan@nwsuaf.edu.cn

*  Correspondence: xuzhenggang@nwafu.edu.cn; Tel.: +86-1868-494-5647

**Abstract:** *Broussonetia papyrifera* is an important ecological and economic tree species. The sexual reproduction of *B. papyrifera* not only has a low germination rate, but also requires high environmental conditions. Therefore, asexual propagation using tissue culture can effectively improve the propagation efficiency of *B. papyrifera*. In this study, the leaves and budded shoots of *B. papyrifera* were used as explants, and different concentrations of plant growth regulators were added to Murashige and Skoog medium (MS) to establish a suitable system for explant callus formation, adventitious buds differentiation and rooting. The results showed that MS + 0.50 mg/L naphthaleneacetic acid (NAA) + 0.25 mg/L 6-benzyladenine(6-BA) and MS + 0.25 mg/L NAA + 0.50 mg/L 6-BA were the best mediums for rapid callus induction from leaf explants and shoot explants, respectively. The best medium combination for shoot differentiation and proliferation was MS + 0.05 mg/L NAA + 0.50 mg/L 6-BA, and the high propagation coefficient could also promote adventitious bud growth. The best rooting medium in the establishment of *B. papyrifera* tissue culture was MS + 0.25 mg/L NAA. Under this condition, the average rooting numbers of leaf explants and shoot explants were 1.71 and 13.86, respectively. In addition, the best transplanting substrate was a mixture of soil:perlite:vermiculite (20:1:1), and the survival rate was 91.1%. This study established a propagation system in vitro culture of *B. papyrifera*, and provided a reference for tissue culture of other woody plants.

**Keywords:** *Broussonetia papyrifera*; MS medium; NAA; 6-BA; differentiation and proliferation; rooting





## 1. Introduction

Plant tissue culture is the fastest and most effective way to produce virus-free plants, which can solve the problem of an insufficient supply of plants. Nowadays, plant tissue culture technology is widely used in various plants to promote factory production of plants and protect endangered plants, especially in woody plants [1,2]. Moreover, tissue culture technology can preserve the fine genetic traits of plants [3]. Plant growth regulators play a very important role at various stages in plant tissue culture [4]. In the rooting experiment of *Coleonema album* Thunb. explants, the rooting rate of 5.00 μM indolebutyric acid (IBA) was 42.5% higher than that without any plant growth regulators [5]. The concentration of growth regulators was also important for plant growth [6]. In an experiment on the effect of plant growth regulators concentration on the rooting of *Magnolia lucida* B. L. Chen & S. C. Yang explants, the rooting rate of 0.60 mg/L naphthaleneacetic acid (NAA) and 1.00 mg/L IBA was significantly higher than that of other combinations [7]. Besides, the choice of explants is also an important factor. The study found that the callus induction rate of hypocotyls, roots, stem cotyledons and stem segments of *Fraxinus mandshurica* Rupr. was different, and the effect of stem cotyledon was the best [8]. The mature seeds of four Malaysian upland rice (*Oryza sativa* L.) cultivars, namely Kusan, Lamsan, Selasi



and Siam, were used as explants to induce callus and plant regeneration and found that Lamsan and Selasi were the most applicable upland rice cultivars to produce high yielding quality crop [9]. It could be seen that the selection of suitable explants and the concentration combination of plant growth regulators was very important for plant growth and development.

*Broussonetia papyrifera*, a tall deciduous tree with wide distribution and strong reproduction ability, has high economic and ecological value. Different parts of *B. papyrifera* were used in industry and medicine [10]. In addition, *B. papyrifera* has a well-developed root system, which could highly adapt to adverse environments such as drought [11] and heavy metal stress [12,13], and plays an important role in ecological restoration [14]. In recent year, the functional development of *B. papyrifera* has gained increasing attention. In addition to wild *B. papyrifera*, hybrid *B. papyrifera* (*B. kazinoki* x *B. papyrifera*) varieties were also bred [12,14]. The propagation of *B. papyrifera* mainly rely on on seed propagation and asexual cutting, but the survival rate is low [15,16]. In order to ensure the quality of *B. papyrifera* is more consistent, many scholars have used tissue culture to cultivate high-yield and high-quality *B. papyrifera* plantlets [17,18], especially for hybrid *B. papyrifera*. Among them, through the culture and expansion of the shoots of the hybrid *B. papyrifera* in vitro, the effects of different hormones on callus, adventitious bud differentiation and rooting were studied [19]. In addition, Wei et al. used hybrid *B. papyrifera* leaves as explants and established an effective regeneration system based on proper cultivation [20].

The genetic background of wild *B. papyrifera* and hybrid *B. papyrifera* is different. While there were a few research papers on tissue culture of wild *B. papyrifera*, wild *B. papyrifera* is widely distributed and rich in genetic diversity. It is necessary to establish an efficient tissue culture system for wild *B. papyrifera* and it is the basis for the development of *B. papyrifera* industry. In this study, leaves and shoots were both selected as explants. NAA and 6-benzyladenine (6-BA) were used as plant growth regulators. The effects of different concentrations of growth regulators on the callus formation, adventitious bud differentiation and rooting were explored. Then, river sand, campus red loam and a mixture of soil: perlite: vermiculite (20:1:1) were used as transplanting substrates to explore the influence of different substrates on the transplanting effect. The research will to provide high-quality resources for the factory production of wild *B. papyrifera*.

## 2. Materials and Methods

### 2.1. Plant Materials

The *B. papyrifera* explants were obtained through seed germination, the seeds for which were collected from Central South University of Forestry and Technology, Changsha, China (E: 112°59′40.2″, N: 28°8′4.2″). All the seeds were collected from the same tree. Soaked the *B. papyrifera* seeds in 75% ethanol for about 30 s and in 0.1% $HgCl_2$ solution for 5-6 minutes, then rinsed them with sterile water 5–6 times. Finally, the seeds were put into the auxin-free Murashige and Skoog (MS) medium [21] to germinate. After the wild *B. papyrifera* plantlets grew to 3–5 cm, the leaves and shoots were taken as explants, respectively. The leaves were cut into 1 * 1 cm squares along the main vein and the shoots were cut into 1–2 cm in the ultra-clean workbench (SW-CJ-2F, Suzhou Bolaier Purification Equipment Co., Ltd. Suzhou, China).

### 2.2. Experimental Design

MS medium was used as the basic medium, the amount of sucrose and agar were added as 25 g/L and 6.5 g/L, respectively. The pH of all mediums were between 5.80–6.00. In order to explore the effect of plant hormones on wild *B. papyrifera* tissue culture, different hormone concentration combinations were designed. Based on different concentration gradients of NAA and 6-BA, which NAA was 0.00, 0.05, 0.10, 0.25, 0.50 mg/L and 6-BA was 0.00, 0.05, 0.10, 0.25, 0.50 mg/L, 25 experimental groups were designed in the research (Table 1). The explants were inoculated in MS medium with different concentrations of NAA and 6-BA. Three explants were inoculated in each tissue culture flask and seven

replicates were set for each experimental group. The medium without plant growth regulators (group 1) was used as a control.

**Table 1.** Combinations of naphthaleneacetic acid (NAA) and 6-benzyladenine (6-BA) in different concentrations.

| Group | Plant Growth Regulators | |
| --- | --- | --- |
| | NAA (mg/L) | 6-BA (mg/L) |
| 1 | 0.00 | 0.00 |
| 2 | 0.00 | 0.05 |
| 3 | 0.00 | 0.10 |
| 4 | 0.00 | 0.25 |
| 5 | 0.00 | 0.50 |
| 6 | 0.05 | 0.00 |
| 7 | 0.05 | 0.05 |
| 8 | 0.05 | 0.10 |
| 9 | 0.05 | 0.25 |
| 10 | 0.05 | 0.50 |
| 11 | 0.10 | 0.00 |
| 12 | 0.10 | 0.05 |
| 13 | 0.10 | 0.10 |
| 14 | 0.10 | 0.25 |
| 15 | 0.10 | 0.50 |
| 16 | 0.25 | 0.00 |
| 17 | 0.25 | 0.05 |
| 18 | 0.25 | 0.10 |
| 19 | 0.25 | 0.25 |
| 20 | 0.25 | 0.50 |
| 21 | 0.50 | 0.00 |
| 22 | 0.50 | 0.05 |
| 23 | 0.50 | 0.10 |
| 24 | 0.50 | 0.25 |
| 25 | 0.50 | 0.50 |

*2.3. Determination of Morphological Index*

The inoculated explants were maintained in a culture room (the temperature and illumination time could be controlled) with an illumination time of 12 h/d and a temperature of $25 \pm 2$ °C. We set the light intensity to 2000 lx. The callus, differentiation, proliferation, and rooting situation were observed everyday. After 60 d, we recorded the length and width of the callus, the number of roots and adventitious buds in the leaf groups (Figure 1a1–a3). We recorded the diameter of the callus, the number of roots, branches and leaves, the length of the longest root, and the height of the longest shoot in the shoot groups (Figure 1b1–b3).

*2.4. Acclimatizing and Transplanting of Tissue Culture Plantlets*

The suitable medium for rooting was selected through the experiment of different plant growth regulators combinations on the number of roots of *B. papyrifera* explants. The *B. papyrifera* shoots were obtained after seed germination and were cut into 1–2 cm sections on an ultra-clean workbench, then inoculated on the rooting medium. After rooting induction culture, tissue culture plantlets of *B. papyrifera* could be opened and acclimatized when the height was about 4 cm and the root system was strong, so that they could gradually adapt to the culture room environment. The mouth of the tissue culture flask was covered with a plastic film and poked several small holes. The temperature was maintained at 25 °C. After 5 d, the plastic film was removed to further strengthen the contact with the culture room environment. The plantlets were taken out after 5 d of further cultured, the culture medium on the roots was washed, and 0.1% carbendazim solution was used for disinfection and sterilization. Then the tissue culture plantlets transplanted to the substrate of river sand, campus red loam and a mixture of soil: perlite: vermiculite

(20:1:1), respectively. Each matrix transplanted 30 plantlets in the culture room with an illumination time of 12 h/d (the light intensity is 2000 lx) and a temperature of 25 ± 2 °C, repeat three times, the nutrient solution (dissolve 1.26 g Hoagland nutrient solution and 0.945 g calcium salt in 1000 mL distilled water) was sprayed every 7 d (30 mL per sprayed), and the survival rate after 45 d was counted and calculated.

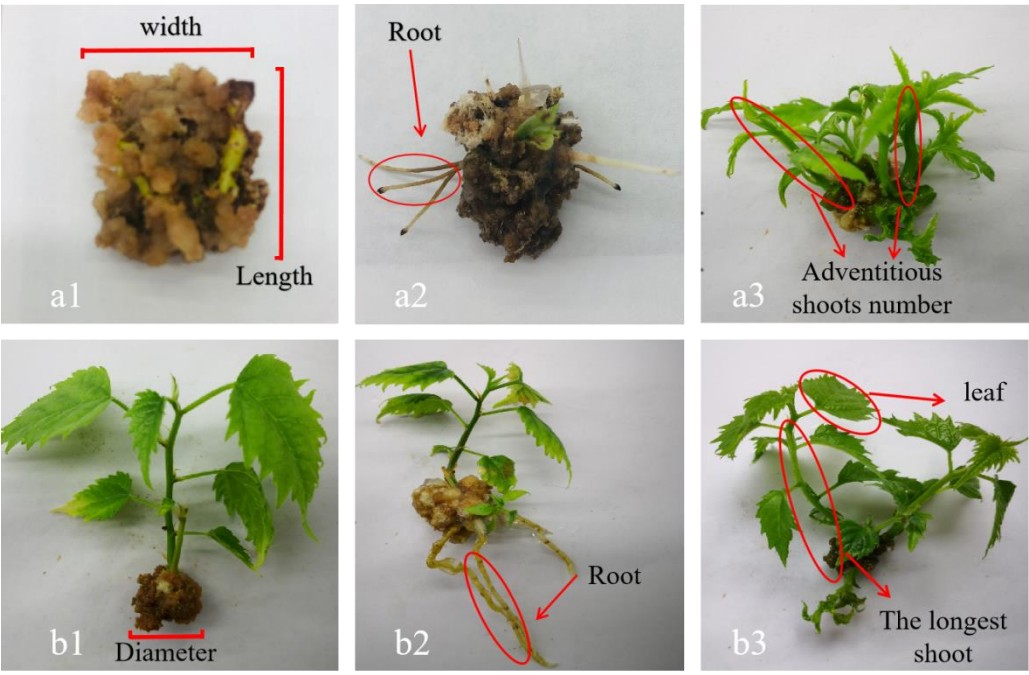

**Figure 1.** Determination of *B. papyrifera* explants. (**a1**) Callus of leaf explant; (**a2**) rooting of leaf explants; (**a3**) the adventitious buds differentiation of leaf explants; (**b1**) callus of shoot explants; (**b2**) rooting of shoot explants; (**b3**) the adventitious buds differentiation of shoot explants.

### 2.5. Statistical Analysis

The experiment was designed completely at random. The SPSS 20.0 software was used for two-way analysis of variance, and Duncan's multiple range test was used to test the statistically significant differences ($p < 0.05$). In addition, the SPSS 20.0 software was also used for a non-parametric test, and the Kruskal–Wallis test was used to test statistically significant differences ($p < 0.05$). The Wilcox test in R was used to calculate multiple pairwise comparisons between groups for the non-parametric test. Sigmaplot 12.5 software was used to draw statistical graphs.

### 3. Results

### 3.1. Growth of Wild B. papyrifera Explants

Twenty five groups of plant growth regulators with different concentrations all promoted the growth of shoot and leaf explants. In leaf explants, the callus gradually formed at the incision in 15–20 d (Figure 1a1,a2). While the callus in shoot explants were formed in 10–15 d (Figure 2b1,b2). The callus and adventitious buds formed in shoot explants were earlier and faster than the leaf explants. After 30–40 d, the adventitious buds in leaf explants were formed (Figure 2a3), while in shoot explants they were formed only in 15–20 d (Figure 2b2). During the entire growth process of explants, only a small part of them could not form adventitious buds. At about 20 d, rooting only appeared in a few explants. At harvest (60 d), the growth rate in shoot explants were better than that of leaf explants, while the degree of differentiation in shoot explants were worse than that of leaf explants (Figure 2a4,b4).

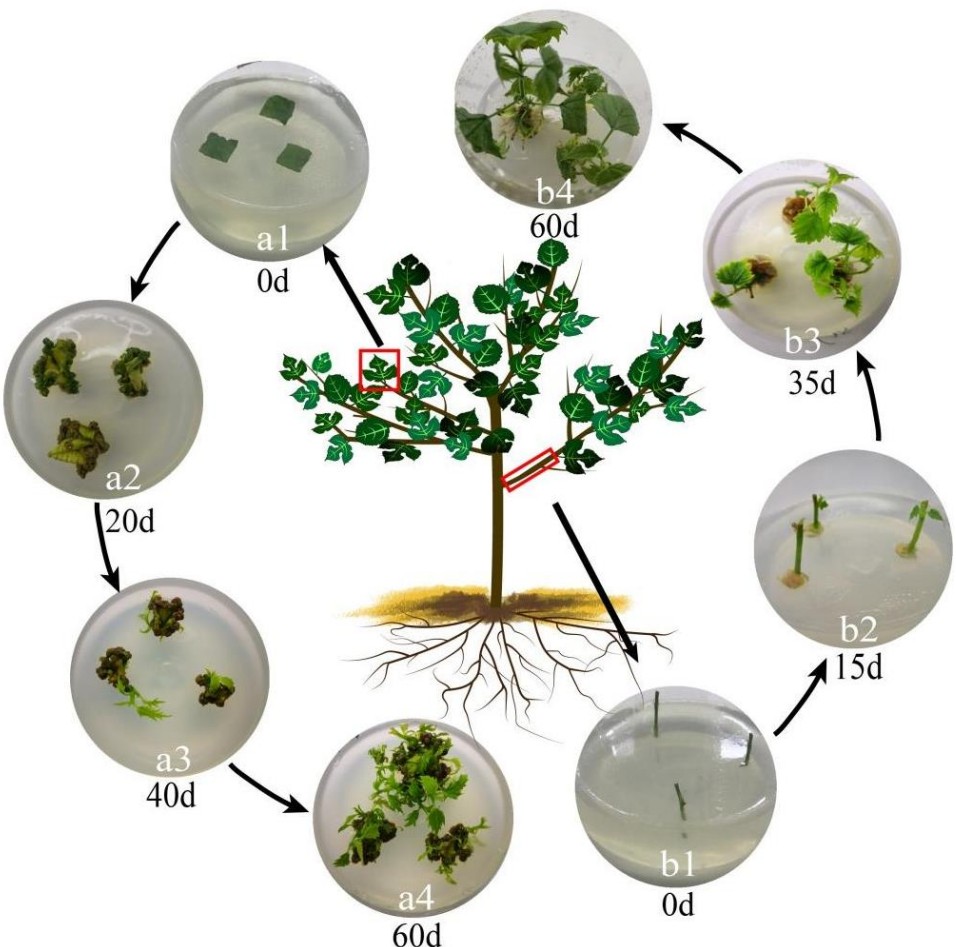

**Figure 2.** Schematic diagram of the growth of *B. papyrifera* explants. (**a1**–**a4**) The growth process of leaf explants; (**b1**–**b4**) the growth process of shoot explants.

### 3.2. Induction of Explant Callus

According to the results of the two-way analysis of variance, the interaction between NAA and 6-BA has a significant effect on the area of leaf callus ($p < 0.001$) and the diameter of shoot callus ($p < 0.01$). In the 25 experimental groups, the areas of callus were not consistent. Without adding plant growth regulators, leaf explants could not form callus and shoot explants only formed little callus. With the increase of plant growth regulators concentration, the areas of leaf explants callus and the diameters of shoot explants callus were all increased, indicating that plant growth regulators could induce the formation of callus. Under the same NAA concentration, the change of the callus area of most leaf explants increased as the concentration of 6-BA increased (Table 2). When the 6-BA concentration reached 0.25 mg/L and 0.50 mg/L, the area of the leaf explant callus was generally higher than other levels. As the NAA concentration increased, leaf explants could better form the callus. The callus area of leaf explants reached a maximum of $2.73 \pm 0.30$ cm$^2$ when the NAA concentration was 0.50 mg/L and the 6-BA concentration was 0.25 mg/L. The change of callus diameter of shoot explants was similar to that of leaf explants (Table 3), and the overall trend was that it increased with the increase of 6-BA concentration. The results showed that the diameter of shoot explants callus reached the maximum value of $1.3 \pm 0.15$ cm when the NAA concentration was 0.25 mg/L and the 6-BA concentration was 0.50 mg/L.

### 3.3. Induction of Explants Adventitious Buds

According to the results of non-parametric tests, the interaction between NAA and 6-BA had a significant impact on the number of adventitious buds differentiation ($p < 0.001$) of the two explants. The induction of adventitious buds could reflect the ability of differentiation and proliferation. The initial values of adventitious buds for leaf explants and shoot explants were 0 and 1, respectively (Figure 2a1,b1). Without the addition of 6-BA, the leaf explants could not differentiate to form adventitious buds, and only a small part of the shoot explants could differentiate into adventitious buds (Tables 4 and 5). It could be seen that 6-BA has a significant effect on the induction of adventitious buds. When the concentration of 6-BA was 0.00, 0.05 and 0.10 mg/L, the change of NAA concentration had little effect on the differentiation of adventitious buds, and the degree of adventitious buds differentiation of explants was low (Tables 4 and 5). As the concentration of 6-BA increased to 0.25 mg/L, the adventitious buds of leaf explants and shoot explants had obvious differentiation. Under the condition of 0.50 mg/L 6-BA concentration and 0.50 mg/L NAA concentration, the number of adventitious buds of leaf explants reached the highest degree of differentiation and the average number of adventitious buds was $10.14 \pm 1.82$ (Table 4). The number of adventitious buds in shoot explants reached the highest differentiation when NAA concentration was 0.05 mg/L and 6-BA was 0.50 mg/L, the average number of adventitious buds was $5.33 \pm 1.19$ (Table 5).

**Table 2.** Effects of different plant growth regulators combinations on callus area ($cm^2$) of *B. papyrifera* leaf explants.

| 6-BA (mg/L) | NAA (mg/L) | | | | |
|---|---|---|---|---|---|
| | 0.00 | 0.05 | 0.10 | 0.25 | 0.50 |
| 0.00 | 1.00 b D | $1.57 \pm 0.13$ bc C | $1.76 \pm 0.17$ b BC | $2.06 \pm 0.08$ a A | $1.82 \pm 0.08$ b BC |
| 0.05 | $1.15 \pm 0.08$ b B | $1.83 \pm 0.15$ ab A | $1.62 \pm 0.17$ b AB | $2.07 \pm 0.26$ a A | $1.93 \pm 0.18$ b A |
| 0.10 | $1.94 \pm 0.19$ a AB | $2.31 \pm 0.24$ a A | $2.00 \pm 0.15$ ab AB | $1.75 \pm 0.16$ a B | $2.23 \pm 0.18$ a AB |
| 0.25 | $1.93 \pm 0.19$ a AB | $1.27 \pm 0.10$ c B | $2.18 \pm 0.23$ ab AB | $2.28 \pm 0.33$ a AB | $2.73 \pm 0.30$ a A |
| 0.50 | $1.19 \pm 0.08$ b B | $1.99 \pm 0.16$ ab AB | $2.43 \pm 0.20$ a A | $2.13 \pm 0.26$ a A | $2.70 \pm 0.30$ a A |

Note: Data statistics at harvest (60 d). Each value represents the mean $\pm$ standard error of multiple replicate samples. Different letters in the same column or row indicate significant difference ($p < 0.05$) by Duncan's multiple range test. Lowercase letters indicate the significance of the 6-BA, and uppercase letters indicate the significance of the NAA.

**Table 3.** Effects of different plant growth regulators combinations on callus diameter (cm) of *B. papyrifera* shoot explants.

| 6-BA (mg/L) | NAA (mg/L) | | | | |
|---|---|---|---|---|---|
| | 0.00 | 0.05 | 0.10 | 0.25 | 0.50 |
| 0.00 | $0.29 \pm 0.04$ b C | $0.55 \pm 0.08$ b BC | $0.58 \pm 0.03$ b BC | $0.84 \pm 0.12$ b B | $1.10 \pm 0.15$ a A |
| 0.05 | $0.32 \pm 0.06$ b C | $0.95 \pm 0.08$ a B | $1.00 \pm 0.17$ a B | $1.08 \pm 0.04$ ab AB | $1.27 \pm 0.09$ a A |
| 0.10 | $0.22 \pm 0.04$ b B | $0.97 \pm 0.15$ a A | $1.07 \pm 0.08$ a A | $1.19 \pm 0.07$ a A | $0.99 \pm 0.20$ a A |
| 0.25 | $0.49 \pm 0.05$ a C | $0.98 \pm 0.10$ a AB | $1.04 \pm 0.11$ a AB | $0.83 \pm 0.12$ ab B | $1.17 \pm 0.19$ a A |
| 0.50 | $0.60 \pm 0.11$ a B | $0.91 \pm 0.13$ a AB | $1.04 \pm 0.07$ a AB | $1.30 \pm 0.15$ a A | $1.10 \pm 0.15$ a AB |

Note: Data statistics at harvest (60 d). Each value represents the mean $\pm$ standard error of multiple replicate samples. Different letters in the same column or row indicate significant difference ($p < 0.05$) by Duncan's multiple range test. Lowercase letters indicate the significance of the 6-BA, and uppercase letters indicate the significance of the NAA.

### 3.4. Induction of Explants Rooting

The interaction between NAA and 6-BA also had a significant impact on the number of roots ($p < 0.001$) of the two explants. The combination of suitable plant growth regulators could promote root formation (Figure 3). When the concentration of NAA was low, no matter how the concentration of 6-BA changed, the rooting effect of leaf and shoot explants was not obvious (Tables 6 and 7). With the increase of NAA concentration, the rooting ability of explants gradually strengthened. When the NAA concentration increased to 0.25 mg/L, the rooting effect of explants was the best, and the number of rooting explants was significantly increased. At this concentration (0.25 mg/L NAA, 0.00 mg/L 6-BA), leaf

and shoot explants reached the maximum number of roots, and the average number of roots were 1.71 ± 1.04 and 13.86 ± 3.49, respectively (Tables 6 and 7).

**Table 4.** Effects of different plant growth regulators combinations on the number of adventitious buds of *B. papyrifera* leaf explants.

| 6-BA (mg/L) | NAA (mg/L) | | | | |
|---|---|---|---|---|---|
| | 0.00 | 0.05 | 0.10 | 0.25 | 0.50 |
| 0.00 | 0.00 b A | 0.00 b A | 0.00 c A | 0.00 b A | 0.00 c A |
| 0.05 | 0.00 b B | 0.00 b B | 0.00 c B | 0.33 ± 0.33 ab AB | 1.33 ± 0.5 b A |
| 0.10 | 0.4 ± 0.34 b A | 0.43 ± 0.34 b A | 0.22 ± 0.15 c A | 0.67 ± 0.29 ab A | 0.43 ± 0.30 bc A |
| 0.25 | 1.80 ± 0.67 a B | 3.67 ± 1.34 a AB | 4.39 ± 1.25 a A | 2.44 ± 1.36 a AB | 2.43 ± 1.62 b AB |
| 0.50 | 0.00 b C | 3.00 ± 0.87 a B | 1.50 ± 0.40 b B | 0.00 b C | 10.14 ± 1.82 a A |

Note: Data statistics at harvest (60 d). Each value represents the mean ± standard error of multiple replicate samples. The row and the column variables are statistically significantly associated at $p < 0.001$, by Kruskal–Wallis test for count data. Different letters in the same column or row indicate significant difference ($p < 0.05$) by Wilcox test. Lowercase letters indicate the significance of the 6-BA, and uppercase letters indicate the significance of the NAA.

**Table 5.** Effects of different plant growth regulators combinations on the number of adventitious buds of *B. papyrifera* shoot explants.

| 6-BA (mg/L) | NAA (mg/L) | | | | |
|---|---|---|---|---|---|
| | 0.00 | 0.05 | 0.10 | 0.25 | 0.50 |
| 0.00 | 1.00 c B | 1.2 ± 0.11 b A | 1.00 b B | 1.00 b A | 1.38 ± 0.18 a A |
| 0.05 | 1.00 c C | 1.58 ± 0.34 b B | 1.75 ± 0.37 a AB | 2.44 ± 1.49 a A | 1.44 ± 0.24 a B |
| 0.10 | 1.40 ± 0.24 b A | 2.17 ± 0.67 b A | 2.28 ± 0.42 a A | 1.93 ± 0.37 ab A | 1.57 ± 0.30 a A |
| 0.25 | 1.40 ± 0.16 b AB | 4.00 ± 1.03 ab A | 4.15 ± 1.12 a A | 1.56 ± 0.34 ab AB | 1.14 ± 0.14 a B |
| 0.50 | 2.11 ± 0.26 a AB | 5.33 ± 1.19 a A | 1.57 ± 0.17 a B | 1.11 ± 0.11 b B | 2.00 ± 0.69 a AB |

Note: Data statistics at harvest (60 d). Each value represents the mean ± standard error of multiple replicate samples. The row and the column variables are statistically significantly associated at $p < 0.001$, by Kruskal–Wallis test for count data. Different letters in the same column or row indicate significant difference ($p < 0.05$) by Wilcox test. Lowercase letters indicate the significance of the 6-BA, and uppercase letters indicate the significance of the NAA.

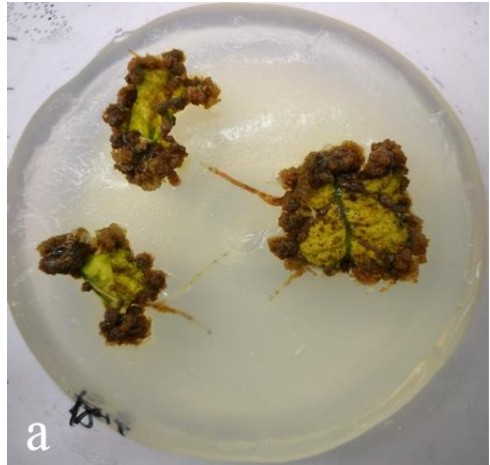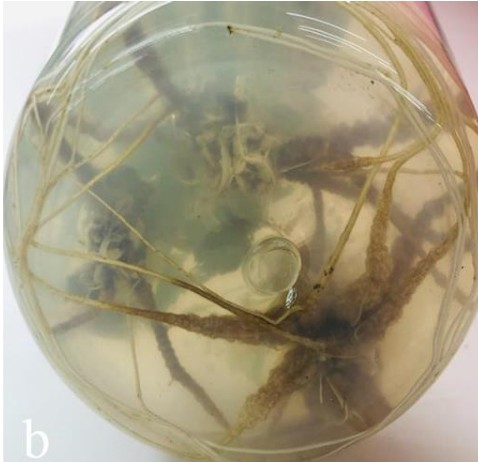

**Figure 3.** Induced rooting of *B. papyrifera* explants. (**a**) Rooting of leaf explant; (**b**) rooting of shoot explants.

**Table 6.** Effects of different plant growth regulators combinations on the number of roots of *B. papyrifera* leaf explants.

| 16-BA (mg/L) | NAA (mg/L) | | | | |
|---|---|---|---|---|---|
| | 0 | 0.05 | 0.1 | 0.25 | 0.5 |
| 0.00 | 0.00 a B | 0.00 a B | 0.10 ± 0.10 ab AB | 1.71 ± 1.04 a A | 0.63 ± 0.50 a AB |
| 0.05 | 0.00 a B | 0.00 a B | 1.25 ± 0.56 a A | 0.00 b B | 0.67 ± 0.55 a AB |
| 0.10 | 0.00 a B | 0.00 a B | 0.00 b AB | 0.00 b B | 0.27 ± 0.18 a A |
| 0.25 | 0.00 a A | 0.08 a A | 0.00 b A | 0.00 b A | 0.00 a A |
| 0.50 | 0.00 a B | 0.00 a B | 0.00 b B | 1.11 ± 0.26 a A | 0.00 a B |

Note: Data statistics at harvest (60 d). Each value represents the mean ± standard error of multiple replicate samples. The row and the column variables are statistically significantly associated at $p < 0.001$, by Kruskal–Wallis test for count data. Different letters in the same column or row indicate significant difference ($p < 0.05$) by Wilcox test. Lowercase letters indicate the significance of the 6-BA, and uppercase letters indicate the significance of the NAA.

**Table 7.** Effects of different plant growth regulators combinations on the number of roots of *B. papyrifera* shoot explants.

| 6-BA (mg/L) | NAA (mg/L) | | | | |
|---|---|---|---|---|---|
| | 0.00 | 0.05 | 0.10 | 0.25 | 0.50 |
| 0.00 | 0.08 ± 0.08 a B | 0.13 ± 0.09 a B | 1.10 ± 0.77 a B | 13.86 ± 3.49 a A | 0.50 ± 0.50 a B |
| 0.05 | 0.00 a B | 0.33 ± 0.33 a AB | 1.25 ± 0.65 a A | 0.78 ± 0.40 b A | 0.22 ± 0.15 a AB |
| 0.10 | 0.00 a B | 0.00 a B | 0.00 b B | 0.80 ± 0.73 b AB | 2.00 ± 1.31 a A |
| 0.25 | 0.00 a A | 0.00 a A | 0.00 b A | 0.00 b A | 0.00 a A |
| 0.50 | 0.00 a A | 0.00 a A | 0.00 b A | 0.89 ± 0.68 b A | 0.00 a A |

Note: Data statistics at harvest (60 d). Each value represents the mean ± standard error of multiple replicate samples. The row and the column variables are statistically significantly associated at $p < 0.001$, by Kruskal–Wallis test for count data. Different letters in the same column or row indicate significant difference ($p < 0.05$) by Wilcox test. Lowercase letters indicate the significance of the 6-BA, and uppercase letters indicate the significance of the NAA.

### 3.5. The Number of Leaves, the Longest Shoot and the Longest Root of the Shoot Explant

The shoot explants began to grow leaves when they were connected to the medium, and their length was increasing, but the leaf explants need to develop for a period of time before they could grow adventitious buds and new leaves. The changes of different indicators within 0–60 d could be observed well in shoot explants.

When the shoot explants differentiated into adventitious buds, new leaves could also grow (Figure 4), and the number of leaves could also reflect the degree of differentiation. The number of leaves, the longest shoot and the longest root of shoot explants ($p < 0.001$) had a significant effect on the interaction with NAA and 6-BA. The shoot explants had the best differentiation effect when the NAA concentration was 0.05 mg/L (Table 8). The degree of differentiation increased with the increased of 6-BA concentration. The number of leaves differentiated from shoot explants reached the maximum (19.60 ± 4.27) when the 6-BA concentration was 0.50 mg/L. In addition, the length of the shoot could reflect the effect of plant growth regulators on the growth of explants, the average length of the longest shoot was the longest (7.33 ± 1.71 cm) under the condition of 0.25 mg/L NAA concentration and without 6-BA added (Figure 5).

**Table 8.** Effects of different plant growth regulators combinations on leaf number of *B. papyrifera* shoot explants.

| 6-BA (mg/L) | NAA (mg/L) | | | | |
|---|---|---|---|---|---|
| | 0.00 | 0.05 | 0.10 | 0.25 | 0.50 |
| 0.00 | 2.08 ± 0.40 b C | 4.67 ± 0.80 b AB | 3.50 ± 0.40 b B | 5.86 ± 1.22 a A | 5.00 ± 0.66 a AB |
| 0.05 | 2.58 ± 0.53 b B | 4.42 ± 0.84 b AB | 7.00 ± 2.47 ab A | 11.33 ± 4.12 a A | 6.89 ± 1.63 a A |
| 0.10 | 3.27 ± 1.37 b B | 8.75 ± 3.42 ab A | 6.44 ± 1.12 ab A | 6.53 ± 2.60 a A | 2.57 ± 1.11 a B |
| 0.25 | 4.20 ± 0.76 ab A | 15.11 ± 7.22 ab A | 13.08 ± 2.83 a A | 3.44 ± 0.85 a A | 3.14 ± 1.16 a A |
| 0.50 | 8.11 ± 2.15 a AB | 19.60 ± 4.27 a A | 3.86 ± 0.85 b B | 4.33 ± 0.67 a B | 4.00 ± 0.87 a B |

Note: Data statistics at harvest (60 d). Each value represents the mean ± standard error of multiple replicate samples. The row and the column variables are statistically significantly associated at $p < 0.001$, by Kruskal–Wallis test for count data. Different letters in the same column or row indicate significant difference ($p < 0.05$) by Wilcox test. Lowercase letters indicate the significance of the 6-BA, and uppercase letters indicate the significance of the NAA.

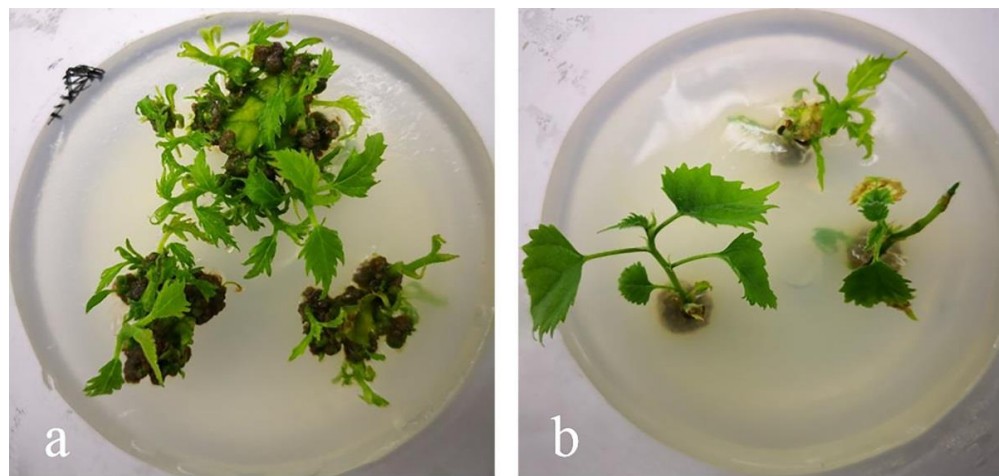

**Figure 4.** Induction of adventitious buds through *B. papyrifera* explants. (**a**) Induction of adventitious buds through leaf explants; (**b**) induction of adventitious buds through shoot explants.

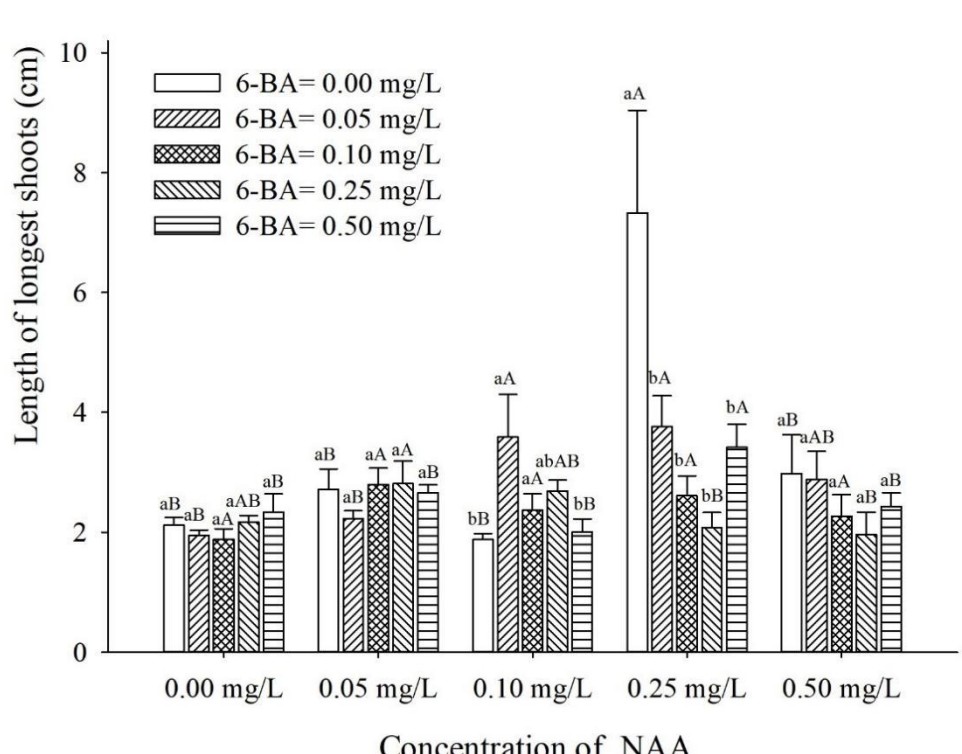

**Figure 5.** Effects of different plant growth regulators combinations on the longest shoot of *B. papyrifera* shoot explants. Data statistics at harvest (60 d). Lowercase letters indicate the significance of the 6-BA, and uppercase letters indicate the Scheme 0. by Duncan's multiple range test. * and *** indicates significant effect at difference statistical level ($p < 0.05$ and $p < 0.0001$ respectively).

Except for the number of roots, the length of roots was also related to the rooting ability. The longest root length of shoot explants had the highest length ($2.67 \pm 0.74$ cm) when NAA concentration was 0.25 mg/L, and 6-BA concentration was 0.00 mg/L (Table 9). According to Tables 7 and 9, it could be seen that the ability of plant growth regulators to induce rooting of explants was NAA > 6-BA, and the rooting effect of explants was better when the NAA concentration reached 0.25 mg/L.

**Table 9.** Effects of different plant growth regulators combinations on the longest root length (cm) of *B. papyrifera* shoot explants.

| 6-BA (mg/L) | NAA (mg/L) | | | | |
|---|---|---|---|---|---|
| | 0.00 | 0.05 | 0.10 | 0.25 | 0.50 |
| 0.00 | 0.02 ± 0.02 a B | 0.63 ± 0.44 a B | 0.10 ± 0.07 b B | 2.67 ± 0.74 a A | 0.28 ± 0.28 a B |
| 0.05 | 0.00 a B | 0.04 ± 0.04 a B | 1.09 ± 0.62 a A | 0.92 ± 0.55 b AB | 0.32 ± 0.23 a AB |
| 0.10 | 0.00 a B | 0.00 a B | 0.00 b B | 0.29 ± 0.23 b AB | 0.50 ± 0.33 a A |
| 0.25 | 0.00 a A | 0.00 a A | 0.00 b A | 0.00 b A | 0.00 a A |
| 0.50 | 0.00 a B | 0.00 a B | 0.00 b B | 0.69 ± 0.48 b A | 0.00 a B |

Note: Data statistics at harvest (60 d). Each value represents the mean ± standard error of multiple replicate samples. Different letters in the same column or row indicate significant difference ($p < 0.05$) by Duncan's multiple range test. Lowercase letters indicate the significance of the 6-BA, and uppercase letters indicate the significance of the NAA.

### 3.6. Transplanting Condition of Tissue Culture Plantlets

The tissue culture plantlets of *B. papyrifera* were cultivated in an environment with superior culture room conditions and MS medium, which resulted in poor root absorption ability. The direct transplantation and great difference between flask environment and culture room conditions led to low survival rate. Therefore, it was necessary to carry out transplanting by steps so that the tissue culture plantlets could gradually adapt to the culture room environmental conditions (Figure 6). Comparing the effects of different plantlets transplanting substrates on the survival rate of tissue culture plantlets, it was found that mixture of soil: perlite: vermiculite (20:1:1) was the best transplanting substrate, with the highest transplanting survival rate reaching 91.1%, followed by campus red loam with a survival rate of 81.1%, and river sand with the lowest survival rate of 62.2% (Table 10).

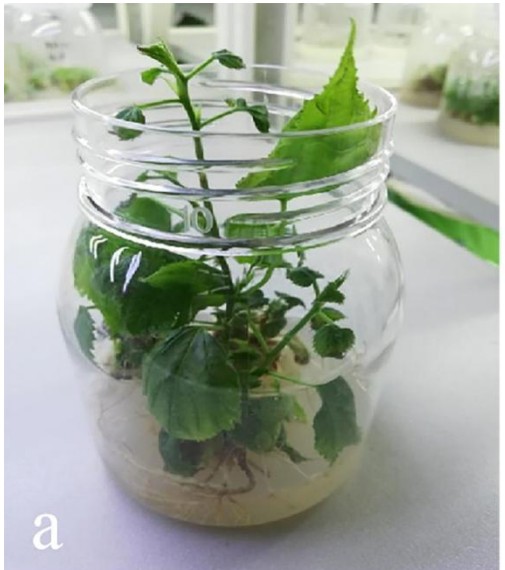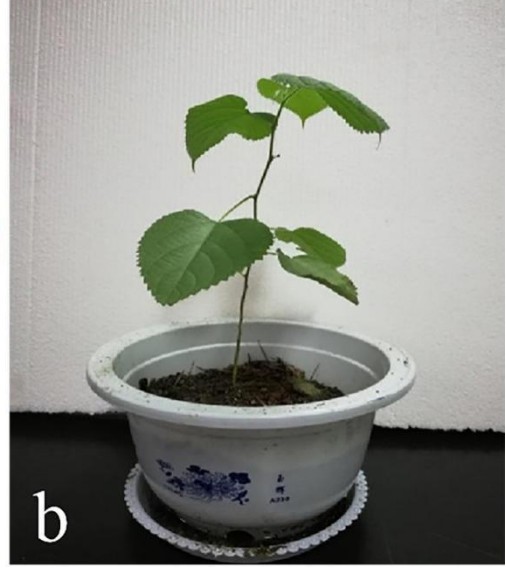

**Figure 6.** Acclimatizing and transplanting of *B. papyrifera* tissue culture plantlets. (**a**) Acclimatizing of tissue culture plantlets; (**b**) growth after three months of transplanting.

**Table 10.** Effects of different substrates on transplanting of *B. papyrifera* tissue culture plantlets.

| Treatment Group | Transplanting Substrate | Number of Surviving Plants | Survival Rate (%) |
|---|---|---|---|
| 1 | River sand | 18.67 ± 1.15 c | 62.2 ± 3.8 c |
| 2 | Campus red loam | 24.33 ± 1.53 b | 81.1 ± 5.1 b |
| 3 | a mixture of soil: perlite: vermiculite (20:1:1) | 27.33 ± 0.57 a | 91.1 ± 1.9 a |

Note: Lowercase letters indicate the significance of the different treatments. Different letters in the same column indicate significant difference ($p < 0.05$) by Duncan's multiple range test.

## 4. Discussion

Auxin and cytokinin can produce synergistic or antagonistic effects to a certain extent [5,22]. Comparing the inductive effects of NAA and 6-BA on *B. papyrifera* explant callus, 6-BA plays a leading role in the induction of callus, and on this basis, the addition of NAA would further promoted the formation of callus, which was consistent with the results of Liu et al. [23] and Huang et al. [24] (Table 11). Generally, each part of the plant has the potential to induce callus, but the differentiation ability was obviously different, the type and concentration of plant growth regulations required were also different [25,26]. This study showed that the effects of 6-BA and NAA on shoot and leaf explants were not completely the same. The callus formation effect was better under the condition of high concentration of 6-BA, and leaf explants require a higher NAA concentration than shoot explants. In addition, different plants require different concentrations of plant growth regulators. In the study of hemp callus, the most suitable concentration for inducing callus was: 2.00 mg/L 6-BA and 0.50 mg/L NAA [27]. The optimal concentration for *Lilium lancifolium* Thunb. callus induction was: 1.00 mg/L 6-BA + 0.15 mg/L NAA + 0.20 mg/L kinetin (KT) + 0.05 mg/L 2, 4-dichlorophenoxyacetic acid (2,4-D) [28]. While the 6-BA concentrations used in other studies were relatively high, whether a high concentration of 6-BA could further promote the growth of *B. papyrifera* explants remains to be verified by further experiments.

The differentiation and proliferation of adventitious buds is the key to achieving rapid propagation of plant tissue culture [29]. There are many factors that affect the induction of adventitious buds, such as the effect of different hormones, carbon sources, gelling agents and nutrients [30,31]. Among them, the type and concentration of plant growth regulators are the most important factors [32]. 6-BA has a significant effect on the differentiation of adventitious buds of explants, and NAA promotes the growth of adventitious buds. The combined effect of 6-BA and NAA could better promote adventitious buds differentiation, which was consistent with the research results of Yan et al. [17] and Huang et al. [24] on the adventitious buds differentiation of *B. papyrifera* (Table 11). In addition, the effects of 6-BA and NAA on adventitious buds differentiation of leaf and shoot explants were inconsistent. Under the main influence of 6-BA, leaf explants required high concentrations of NAA to better differentiate adventitious buds, while shoot explants required a small amount of NAA. When NAA concentration was 0.05 mg/L and 6-BA concentration was 0.50 mg/L, the number of adventitious buds and leaves of shoot explants reached the maximum. The research on *Ammopiptanthus mongolicus* Chen f. found that the ratio of cytokinin and auxin determines the differentiation of adventitious buds and roots. When the concentration of cytokinin was less than the concentration of auxin, it was beneficial to promote the differentiation of explant roots. Conversely, it was beneficial to the differentiation of adventitious buds [33]. This view was consistent with the results of our study. In addition, the length of the longest shoot of the shoot explants reflected the growth level of *B. papyrifera* tissue culture plantlets and was greatly affected by the concentration of NAA. The results showed that the optimal medium for the longest shoot of the shoot explants was: MS + 0.25 mg/L NAA. Studies on the shoot explants of *Artemisia selengensis* sp. showed that low concentrations of 6-BA and NAA (both were 0.02 mg/L) were beneficial to the elongation of adventitious buds [34]. Conversely, high concentrations could inhibit the growth of buds.

In the research on rooting of plant explants, the factors affected the rooting of explants were not only the type and concentration of plant growth regulators [35], but also the concentration of inorganic salts, the type of medium and so on [30,36]. In this work, the best combination of medium and plant growth regulators for rooting of *B. papyrifera* explants was: MS + 0.25 mg/L NAA. In this condition, the leaf explants and shoot explants reached the maximum number of roots, and the length of shoot explant root was the longest. It could be seen that the main factor that promotes the rooting of *B. papyrifera* explants was the concentration of NAA, which could induce the proliferation and growth of explant roots, but high concentration of NAA could inhibit the growth and rooting of explants. Zhi et al. [18] also obtained similar results in research on *B. papyrifera* rooting, and

the best medium was: 1/2 MS + 0.10 mg/L NAA (Table 11). In the rooting research of other plants, the best medium for rooting of *Dracocephalum rupestre* Hance explants was: MS + 0.10 mg/L NAA [37]. In addition, the best medium that affects the rooting of *Polygonatum cyrtonema* Hua explants was: 1/2 MS + 1.00 mg/L NAA [38]. These results were similar to the results of this study. In previous studies, we also found that 1/2 MS medium has a better rooting effect on plants. In order to facilitate a better comparison and analysis, we uniformly used MS medium. In further experiments, the effect of MS and 1/2 MS medium on the differentiation and rooting of *B. papyrifera* explants will be discussed in order to identify a more suitable tissue culture system.

**Table 11.** Research results of *B. papyrifera* tissue culture.

| Types of *B. papyrifera* | Explants | Time of Each Stage | Optimal Combination of Plant Growth Regulators | References |
|---|---|---|---|---|
| Hybrid | Leaves | Callus appeared in 15 d | Rooting: MS + 2.50 mg/L 6-BA + 2.00 mg/L NAA | [17] |
| Hybrid | Shoots | Statistics of callus on the 30th d | Callus induction: MS + 0.50 mg/L IBA + 2.00 mg/L 6-BA | [19] |
| | | Statistics of adventitious buds on the 30th d | Adventitious buds induction: MS + 1.50 mg/L IBA + 2.00 mg/L 6-BA | |
| | | Rooting in 15 d | Rooting: 1/2 MS + 0.80 mg/L IBA | |
| Hybrid | Leaves | Adventitious buds began to appear after two weeks | Adventitious buds induction: MS + 2.00 mg/L 6-BA + 0.10 mg/L NAA | [20] |
| Hybrid | Leaves | Callus appeared in 12 d | Callus induction: MS + 1.50 mg/L 6-BA + 1.00 mg/L NAA | [23] |
| | | Rooting in 15 d | Rooting: 1/2 MS + 0.50 mg/L 6-BA + 1.00 mg/L NAA | |
| Hybrid | Leaves | Callus appeared in 20 d and Shoot-buds appeared in 30 d | Callus induction and Adventitious buds induction: MS + 2.00 mg/L 6-BA + 0.10 mg/L NAA | [24] |
| | | Statistics of rooting on the 30th d | Rooting: B5 + 0.01 mg/L 6-BA + 0.30 mg/L NAA + 0.50 mg/L IBA | |
| Wild | Shoots | Statistics of rooting on the 35th d | Rooting: 1/2 MS + 0.10 mg/L NAA | [18] |
| Wild | Leaves | Adventitious buds appeared in 3 weeks | Adventitious buds induction: MS + 2.00 mg/L 6-BA + 0.05 mg/L IBA | [39] |

Different transplanting substrates have different survival rates for tissue culture plantlets. Campus red loam was a typical soil type in southern China, with high viscosity and poor water permeability, which leads easily to local water accumulation and root rot. The vermiculite was stable in character and has good air permeability and water permeability, as well as water absorption and retention. Perlite can effectively improve soil porosity, air circulation and plant root elongation [40]. A mixture of soil:perlite:vermiculite (20:1:1) could make the transplanting survival rate reach more than 91%, and the plants grew well and prospered.

**5. Conclusions**

In this study, the leaves and shoots of *B. papyrifera* were used as explants to investigate the effects of different concentrations of plant growth regulators. Through experiments, the best system suitable for callus formation, adventitious buds differentiation and rooting of *B. papyrifera* tissue culture plantlets were established. The results showed that NAA has a greater impact on the formation of callus, the number of roots, the length of shoots and roots; 6-BA has a greater influence on the number of adventitious buds and leaves

differentiated from explants. The optimal media for callus formation of leaf explants and shoot explants were: MS + 0.50 mg/L NAA + 0.25 mg/L 6-BA, MS + 0.25 mg/L NAA + 0.50 mg/L 6-BA. The optimal media for adventitious buds differentiation of leaf explants and shoot explants were: MS + 0.50 mg/L NAA + 0.50 mg/L 6-BA, MS + 0.05 mg/L NAA + 0.50 mg/L 6- BA. The optimal medium for rooting of leaf explants and shoot explants was: MS + 0.25 mg/L NAA. In addition, the best transplanting substrate was a mixture of soil:perlite:vermiculite (20:1:1), and the survival rate was 91.1%. This experiment provided a tissue culture system of *B. papyrifera* that could improve reproduction efficiency and breed excellent germplasm resources.

**Author Contributions:** Conceptualization, J.Z., Z.X. and L.W.; methodology, L.W. and Y.L.; software, J.Z.; validation, G.Y.; formal analysis, J.Z.; data curation, W.Z.; writing—original draft preparation, J.Z.; writing—review and editing, J.Z. and Z.X.; supervision, Y.Z.; project administration, Z.X.; funding acquisition, Y.Z. and Z.X. All authors have read and agreed to the published version of the manuscript.

**Funding:** This research was funded by Natural Science Foundation of Hunan Province, grant number 2019JJ50027, Key Projects of National Forestry and Grassland Bureau, grant number 201801, Forestry Science and Technology Project of Hunan Province, grant number XLK201920 and China Postdoctoral Science Foundation, grant number 2020M683592. and The APC was funded by Key Projects of National Forestry and Grassland Bureau, grant number 201801.

**Institutional Review Board Statement:** Not applicable.

**Informed Consent Statement:** Informed consent was obtained from all subjects involved in the study.

**Data Availability Statement:** Within a reasonable range, the corresponding author can be requested to provide original data.

**Conflicts of Interest:** The authors declare no conflict of interest.

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
