# Peer review of "Effects of Plant Growth Regulators on the Rapid Propagation System of Broussonetia papyrifera L. Vent Explants"

_forests, doi:10.3390/f12070874_

Round 1
Reviewer 1 Report
Main note to the text:
- The stem is not the same as the shoot (the shoot has nodes, internodes, etc.). This should be properly corrected throughout the text and under the figure.
a3 - The adventitious buds differentiation of leaf explant – please check buds to shoots
b3 - the photo shows the longest shoot (it has nodes, leaves, etc.) and is labeled as the stem – please check stem to shoots
- No homogeneous groups (statistical analysis) with the results in tables 4-9 - this should be completed.
Abstract:
All abbreviations should be expanded (the first time they are used in the text) so that the reader knows from the beginning what they mean (e.g. BA, MS). Please do it.
Line 16: budded stems: What is going on? I think it is unfortunate term. Please check and change.
- Materials and Methods:
Line 83: 75% alcohol - was it ethanol? Add
Line 85: MS medium - if it has not been modified, please provide a reference to the literature (no source) Add
Line 96: 6-benzylamino adenine (6-BA) change to 6-Benzyladenine
Line 105: after putting explants on the medium, the flasks were not in the phytotron but in the greenhouse? how did you control the parameters there (temperature and photoperiod)? how were the flasks secured against contamination under such conditions?
Line 116: what was the composition of the rooting medium? add details, including the content of growth regulators
Line 123: why were seedlings sterilization again before being dropped off?
Line 126: …the nutrient 126 solution was sprayed every 7 d… - what is going on? MS sprayed the seedlings during acclimatization? it needs to be clarified and / or redrafted
Results:
Table 4-Table 9: why are the homogeneous groups missing in this tables? please complete the results (it should be presented in the same way as in tables 2 and 3)
References:
- standardize and check the entire reference list (e.g. record names of journals or titles once Caps lock, then not; once there are the initials of the names [16], then the whole names [10])
- a lot of errors and omissions in the literature list e.g. in item 19 there is no number, it should be 58 (9): 120-123
- everything needs to be carefully checked and corrected
Author Response
Dear Reviewer:
Thank you for your comments concerning our manuscript entitled “Effects of plant growth regulators on the rapid propagation system of Broussonetia papyrifera explants” (forests-1237364). Those comments are all valuable and very helpful for revising and improving our paper, as well as the important guiding significance to our researches. We have studied comments carefully and have made correction which we hope meet with approval. Revised portion are marked in red in the paper.

Reviewer 2 Report
The manuscript “Effects of plant growth regulators on the rapid propagation system of Broussonetia papyrifera explants” attempts to establish an efficient system for in vitro mass propagation B. papyrifera using leaf and stem explants. The authors investigated the impact of various levels and combinations of naphthalene acetic acid (NAA) and 6-benzylamino adenine (6-BA) on callus induction, shoot regeneration, and rooting of B. papyrifera. They further studied the effects of substrates on the survival rate of in vitro raised plantlets. However, they failed to show genetic uniformity among the regenerated plants. In addition, the information on the rooting of regenerated shoots is missing. The manuscript is understandable; however, English must be carefully revised with a subject expert.
Detailed comments are listed as follows:
L38-50: There are several reports available on this genus and family members. Please include the available information on the effect of explants and plant growth regulators on the callus, shoot induction, and rooting of the same or related plant species.
L74: Please indicate the name of the “growth regulators” used in this study. Also, justify the selection of these two plant hormones.
L76: nutrient soil? Please let me know the nutrients which are present in vermiculite: perlite.
L105: cultured in a greenhouse? Please indicate the light intensity “illumination.”
L117: tissue culture seedlings?
L117-121: No contamination?
L122: plantlets instead of seedlings.
L126: Please indicate the environmental conditions.
L188: Please provide the data on the rate of adventitious buds differentiation.
L263: rich nutrient medium?
L264: between greenhouse? Figure 6 is not supporting “look like a culture room”.
L356: How many plantlets obtained from a single leaf/stem explant? “B. papyrifera tissue culture seedlings were established”
L358: 6-BA instead of Cytokinin
Author Response
Dear Reviewer,
Thank you for your comments concerning our manuscript entitled “Effects of plant growth regulators on the rapid propagation system of Broussonetia papyrifera explants” (forests-1237364). Those comments are all valuable and very helpful for revising and improving our paper, as well as the important guiding significance to our researches. We have studied comments carefully and have made correction which we hope meet with approval. Revised portion are marked in red in the paper. The main corrections in the paper and the responds to the reviewer’s comments are as flowing:

Round 2
Reviewer 2 Report
Dear authors,
Please address the following comments:
L26-27: a mixture of soil: perlite: vermiculite (20:1:1) instead of “nutrient soil (the mass ratio of nutrient soil: perlite: vermiculite is 20:1:1)”. Please correct it throughout the text.
L61: Delete “seedlings”
L62: Please rewrite “on seed breeding and asexual cutting, but the survival rate of seed breeding”
L64-65: Please rewrite “culture rapid propagation technology to cultivate high-yield, virus-free,
and fast-growing B. papyrifera seedlings”
L67: Please rewrite “effects of different hormones on callus, shoot buds, strong seedlings and roots were”
L69: effective regeneration system? Plant regeneration?
L77: red loam? Delete it and correct the sentence according to materials and methods.
L92: Delete “The explants were cultured in MS differentiation medium for the subsequent experiments” (Refer L100)
L106: maintained in a culture room instead of “cultured in culture room”
L119: Correct it “The suitable medium (MS + 0.25 mg/L NAA) was selected for rooting.” Indicate the size and source of shoots cultured on the rooting medium. Also, indicate the duration of incubation.
L120: plantlets instead of “seedlings” follow throughout the text. Please read the article carefully “Yu, L.; Li, X.; Tian, H.; Liu, H.; Xiao, Y.; Liang, N.; Zhao, X.; Zhan, Y. Effects of Hormones and Epigenetic Regulation on the Callus and Adventitious Bud Induction of Fraxinus mandshurica Rupr. Forests 2020, 11, 590. https://doi.org/10.3390/f11050590”
L282: plantlets instead of “seedlings”
L283: plantlets instead of “seedlings”
L367: plantlets instead of “seedlings”
Author Response
Dear Reviewer,
Thank you for your comments concerning our manuscript entitled “Effects of plant growth regulators on the rapid propagation system of Broussonetia papyrifera explants” (forests-1237364).Those comments are all valuable and very helpful for revising and improving our paper, as well as the important guiding significance to our researches. We tried our best to improve the manuscript and made some changes in the manuscript. These changes will not influence the content and framework of the paper. And here we marked in red in revised paper.We appreciate for your warm work earnestly, and hope that the correction will meet with approval.
Once again, thank you very much for your comments and suggestions.
